# Assessing the Benefits and Harms Associated with Early Diagnosis from the Perspective of Parents with Multiple Children Diagnosed with Duchenne Muscular Dystrophy

**DOI:** 10.3390/ijns10020032

**Published:** 2024-04-15

**Authors:** Oindrila Bhattacharyya, Nicola B. Campoamor, Niki Armstrong, Megan Freed, Rachel Schrader, Norah L. Crossnohere, John F. P. Bridges

**Affiliations:** 1Department of Biomedical Informatics, The Ohio State University College of Medicine, Columbus, OH 43210, USA; oindrila.bhattacharyya@osumc.edu (O.B.); nicola.campoamor@osumc.edu (N.B.C.); john.bridges@osumc.edu (J.F.P.B.); 2Foundation for Angelman Syndrome Therapeutics, Austin, TX 78704, USA; niki.armstrong@cureangelman.org; 3Parent Project Muscular Dystrophy, Washington, DC 20005, USA; megan@parentprojectmd.org (M.F.); rachel@parentprojectmd.org (R.S.); 4Department of Internal Medicine, The Ohio State University College of Medicine, Columbus, OH 43202, USA

**Keywords:** Duchenne muscular dystrophy, newborn screening, early diagnosis

## Abstract

Duchenne muscular dystrophy (DMD) is a rare neuromuscular disorder diagnosed in childhood. Limited newborn screening in the US often delays diagnosis. With multiple FDA-approved therapies, early diagnosis is crucial for timely treatment but may entail other benefits and harms. Using a community-based survey, we explored how parents of siblings with DMD perceived early diagnosis of one child due to a prior child’s diagnosis. We assessed parents’ viewpoints across domains including diagnostic journey, treatment initiatives, service access, preparedness, parenting, emotional impact, and caregiving experience. We analyzed closed-ended responses on a −1.0 to +1.0 scale to measure the degree of harm or benefit parents perceived and analyzed open-ended responses thematically. A total of 45 parents completed the survey, with an average age of 43.5 years and 20.0% identifying as non-white. Younger siblings were diagnosed 2 years earlier on average (*p* < 0.001). Overall, parents viewed early diagnosis positively (mean: 0.39), particularly regarding school preparedness (+0.79), support services (+0.78), treatment evaluation (+0.68), and avoiding diagnostic odyssey (+0.67). Increased worry was a common downside (−0.40). Open-ended responses highlighted improved outlook and health management alongside heightened emotional distress and treatment burdens. These findings address gaps in the evidence by documenting the effectiveness of early screening and diagnosis of DMD using sibling data.

## 1. Introduction

Duchenne muscular dystrophy (DMD) is a rare X-linked genetic disorder affecting 1 in 5000 live male births annually [1,2], with rare occurrences in females (<1/million) [3,4]. DMD results in progressive muscle damage, leading to loss of ambulation in the teen years. This eventually advances to respiratory failure and cardiomyopathy, ultimately causing premature death between the second and fourth decade of life [3,5]. There are eight FDA-approved therapies for DMD, including deflazacort and vamorolone, DMD-specific corticosteroids [6,7], four exon-skipping therapies [8], a recently FDA-approved gene therapy, Elevidys (delandistrogene moxeparvovec-rokl) [9] and Duvyzat, the first nonsteroidal drug [10]. These treatments are likely to be more effective when administered to individuals with limited muscle tissue loss and fibrosis.

Newborn screening (NBS) is a public health service that has the potential to promptly identify many potentially fatal or disabling conditions for which early treatment can result in improved outcomes [11]. This approach holds promise in facilitating timely intervention to mitigate the effect of a condition and maximizing treatment effectiveness, thereby enhancing overall quality of life [12,13,14]. Despite potential benefits of NBS, the implementation of DMD NBS is limited in the US. Notably, New York [15], Minnesota [16], and Ohio [17] plan to have universal screening in 2024, with three pilot studies recently concluded or underway in New York [18], North Carolina [19], and in Boston, Massachusetts [20] to support the inclusion of DMD on the Recommended Uniform Screening Panel (RUSP).

The primary rationale behind advocating for DMD to be included on the RUSP is the potential for enhanced effectiveness of treatments with earlier diagnosis, which can facilitate improved long-term health outcomes [4]. However, due to limited NBS, the actual diagnosis typically occurs around 4–5 years, a timeframe that has remained consistent over the past three decades, despite signs manifesting as early as the first year of life [21,22,23]. This delay in diagnosis can pose significant challenges in terms of hindering adherence to essential standards of care [24,25,26] as well as imposing emotional, psychological, and financial burden, ultimately disrupting overall quality of life [27].

It is difficult to study the benefits or harms of early diagnosis because of the lack of standard practice in diagnosing DMD before age 4. We sought to address this evidence gap by using data on siblings diagnosed with DMD to assess potential benefits and harms associated with various domains of early diagnosis from the perspectives of their parents. These parents experienced early diagnosis in one of their children following a prior diagnosis of their other child—a situation rare among families with multiple children diagnosed with DMD. Such varying ages of diagnosis among siblings impact clinical milestones and treatment access [28], resulting in diverse experiences for their parents. This unique dynamic sets the stage for a natural experiment to explore parents’ lived experiences associated with early diagnosis. This study will provide a more comprehensive understanding of proactive intervention in the early stages of the disease, potentially informing clinicians, researchers, advocacy groups, and policymakers involved in DMD care about the further value of NBS for DMD.

## 2. Materials and Methods

This research was conducted in partnership with Parent Project Muscular Dystrophy (PPMD), an advocacy organization. A web-based survey was used to engage parents or guardians with multiple children diagnosed with DMD to learn about their experiences with early diagnosis of one of their children due to a prior diagnosis of their other children.

### 2.1. Study Participants and Recruitment

Eligible participants were parents or guardians of at least two living children diagnosed with DMD, each at least 18 years old, and residing in the US. The research team identified potential families through the Duchenne Registry, a patient-reported disease registry, that enrolls individuals affected by Duchenne or Becker muscular dystrophy and carrier females [29]. These enrolled participants, including patients aged 18 and above or parents/custodians/legal guardians of children under 18, actively contribute health-related data to the registry, which are stored securely in a HIPAA-compliant database, and provide their consent for deidentified information to be shared with researchers. The study recruitment email was sent twice to registered parents of a living child with a reported diagnosis of DMD. Additionally, study recruitment information was shared in closed Facebook groups specific to parents of children with DMD. Recruitment took place from 20 September 2023 to 12 November 2023, with a $20 Amazon gift card provided as an incentive.

### 2.2. Survey Instruments

The survey collected information on demographic characteristics for parents, including age and race, and their children’s demographic and clinical characteristics, including their ages at diagnosis, their motor functions, receipt of medical therapies, participation in clinical trial, and involvement in support services like Early Intervention Services (EIS), and Individualized Educational Plan (IEP) at school. Parents’ experiences with early diagnosis were measured using both closed- and open-ended questions on their experiences. Closed-ended questions asked parents to assess their lived experiences with early diagnosis across domains including diagnostic journey, treatment initiatives, access to early intervention services, preparedness and expectations, parenting strategy, emotional impact, and caregiving experience. Responses were recorded as “Benefits”, “Harms”, “Neither a benefit nor a harm”, “Both a benefit and a harm”, or “Did not experience”. These experience objects were generated though the real-life experience of the parents and a review of prior research discussing parental experiences with the Duchenne diagnostic odyssey and perspectives on Duchenne NBS [30]. An “Other” open text field was also provided following the closed-ended questions for parents to express additional thoughts. Based on responses to closed-ended questions, the parents were prompted with additional open-ended questions to further elaborate on whether they viewed their experiences as positive or negative effects of early diagnosis (see Appendix A for survey questionnaire).

### 2.3. Data Analysis

We analyzed responses to closed-ended questions by constructing a bidirectional scale and generating standardized scores for parents’ responses to experience objects. This scale aimed to determine if parents perceived their lived experiences of early diagnosis as benefits (coded as +1), harms (coded as −1), or both (coded as 0). We calculated the standardized scores by dividing the counts of responses as benefits, harms, or both for each question on experience by the number of parents who responded to the questions. The scores could have a positive value indicating benefits, a negative value indicating harms, or both a positive and a negative value indicating both benefits and harms. Additionally, we conducted a thematic analysis on the responses to the open-ended questions. We first familiarized ourselves with all responses and identified emerging themes. These potential themes were then organized, refined, and condensed. Direct quotes from families were selected to exemplify each identified theme. We used paired *t*-tests to compare siblings’ diagnostic and clinical data. All analyses, except open-ended questions, were performed in Stata (Stata 18.0, StataCorp LLC, College Station, TX, USA).

## 3. Results

We engaged 45 eligible parents who provided complete responses for all questions related to their lived experiences with early diagnosis. These 45 parents formed the analytic sample, each having two children with DMD. The majority of the siblings in each family were males, with the exception of two families with the youngest children being females, one a carrier and one with dystrophinopathy. The parents had a mean age of 43.5 years, ranging from 32 to 60, with 20.0% identified as non-white.

### 3.1. Cohort Differences

The youngest child received a diagnosis on average 2 years earlier than the oldest (*p* < 0.001), with current mean ages of 13.6 years for the oldest and 10.8 years for the youngest (*p* < 0.001). Overall, 38.7% of the youngest experienced loss of ambulation, compared to 62.9% of the oldest (*p* = 0.003), both by age 10. In addition, 84.4% of the younger siblings initiated DMD-approved therapies compared to 91.1% of the older siblings, with initiation among the youngest occurring one year earlier on average (*p* < 0.001). Of all siblings, 19.5% were on corticosteroids alone (7 out of 36 for youngest and 8 out of 41 for oldest). More younger siblings (27.8%, 10 out of 36) used a combination of corticosteroids, heart medications, and bone health therapies compared to older siblings (21.9%, 9 out of 41). A small percentage of younger siblings (8.3%, 3 out of 36) received gene therapy alongside corticosteroids, heart medications, and/or bone health therapies, compared to 4.8% of older children (2 out of 41). Around 30% of siblings participated in or were currently enrolled in a Duchenne clinical trial by age 7, on average (*p* = 0.396). Among younger siblings, 58.1% accessed EIS (focusing on physical therapy, learning, communications, and emotional skill), by age 1 on average, compared to 52.3% of the older siblings (*p* < 0.001), by age 2 on average. Similar proportions of youngest (79.1%) and oldest (82.2%) siblings had or currently have an IEP at school, by age 6 on average (*p* = 0.263) (Table 1).

There were noticeable variations in siblings’ age of diagnosis. In the majority of families (*n* = 35, 77.8%), the youngest child received a DMD diagnosis significantly earlier than the oldest, with a maximum gap of 5 years. However, some families had no difference in the diagnostic ages of their two children (*n* = 8, 17.8%), and a small number had their older child diagnosed earlier (*n* = 2, 4.4%) (Figure 1).

### 3.2. Parents’ Lived Experience of Early Diagnosis

Most parents found early diagnosis highly beneficial, especially for school preparedness (+0.79) and early access to support services (+0.78). They also valued having more time to evaluate treatment options (+0.68), absence of diagnostic odyssey (+0.67), and increased clinical trial opportunities (+0.59). These aspects were primarily viewed as beneficial, with very few parents expressing mixed feelings. Parents unanimously agreed that early diagnosis facilitated access to medical assistance through the state or Medicaid (+0.48). Other findings were more nuanced. While some parents felt better being prepared (+0.57), others experienced varying degrees of concern (±0.21, −0.02). Prior experience with an older child shaped parents’ expectations for the younger child, evoking both positive reactions (+0.58) and mixed sentiments (±0.18, −0.04). Adjusting parenting style based on past experiences yielded positive results for some parents (+0.53), while others had mixed feelings (±0.07, −0.02). Hope for gene therapy coexisted with uncertainties about its potential outcomes (+0.41, ±0.07, −0.02). While most parents viewed the aspect of earlier initiation of treatments as beneficial (+0.43), some worried about potential side effects (±0.17, −0.05). Parents also had varying experiences regarding the impact of early diagnosis on their ability to bond with their child (+0.26, ±0.02, −0.02). Shorter diagnostic uncertainty (±0.25) and siblings’ shared experiences (±0.21) led to predominantly mixed parental perceptions.

Increased worry (−0.40), in contrast, was the most common downside of early diagnosis. While most parents considered deferred care or having no immediate plan for care as a negative aspect of early diagnosis (−0.11), some valued the extra time to make informed treatment decisions (+0.19). Overall, parents perceived early diagnosis favorably. This is evident from the positive salience values across most domains of early diagnosis, indicating benefits outweighing harms. This is further supported by the statistically significant overall mean of 0.39 (95% CI: 0.30 to 0.46) on the experience scale, signifying a net positive perception of early diagnosis (Figure 2). However, parental perspectives varied based on the difference in diagnostic ages of their children. Parents reported more positive experiences when the younger child received an earlier diagnosis compared to the older child, although this trend was not statistically significant (Figure 3).

Parents’ lived experiences with early diagnosis revealed additional insights into benefits, harms, or both in the “Other” open text field in the survey, which were not addressed in the closed-ended questions. Most parents who provided examples highlighted additional benefits, such as the “ability to evaluate therapies and trials comprehensively before [the] risk of aging out”, and expressed that “my younger son’s school experience was easier, because we already navigated all the hard stuff with the school district with my older son”.

Open-ended questions were used to further explore parents’ perspectives on both the positive and negative effects of early diagnosis, which revealed distinct themes—response, knowledge, planning, treatment, and health for positive effects, and emotional burden, and treatment effects for negative effects. Questions addressing the positive effects of early diagnosis focused on expectations, parenting, school preparation, access to early intervention services, and other general aspects (Table 2). Parents frequently felt empowered to advocate more effectively for their child’s needs. As one parent explained, “knowing my son’s diagnosis helped me to fight for [my daughter]. I knew something was wrong and continued to fight even though everyone dismissed me”. Early diagnosis also enhanced parents’ understanding of available resources, as one parent explained, “we knew what to expect when dealing with a school, what resources we actually had”. Many found early diagnosis beneficial for planning accommodations and adjusting parenting approaches, including adapting accommodations used by the older son, as one parent explained: “because everything I had fought to be put in place for my oldest son was automatically given to my youngest, just tailored to his needs”. Early access to treatments was a major benefit. Parents could start treatment sooner and explore a broader range of treatments. One of the parents explained that “gene therapy has given my children strength and the opportunity to enjoy playing with each other and peers”. They also noted improved health outcomes for younger children compared to older siblings due to early intervention, with one parent explaining, “… my youngest has the abilities to do things that my oldest wasn’t able to from his delayed diagnosis”. Questions addressing the negative aspects of early diagnosis focused on impacts on parenting, parents’ ability to bond with or understand their youngest child, and other general aspects (Table 3). Many parents described the emotional burden associated with a second diagnosis, with one parent expressing that “finding out about their diagnosis pushed me into a state of high anxiety and impacted my ability to be [fully] there for them”. Additionally, parents highlighted the emotional and physical impacts that come with treatments, with one parent explaining that “the harm would be the side effects of the medication. Medication and doctor’s appointments/tests… almost become your life”.

## 4. Discussion

Parents with multiple living children diagnosed with DMD uniquely experienced earlier diagnoses for younger children compared to older ones (in 78.0% of cases). This potentially resulted from screening during the newborn period, possibly facilitated by comprehensive care measures like genetic counselling, carrier testing on mothers, or sibling screening. Younger children were diagnosed 2 years earlier on average than older children, with 27.0% diagnosed within the first year after birth. Such a timely diagnosis holds the potential to assist parents in navigating the historically challenging diagnostic journey that families typically face when seeking a DMD diagnosis [31,32]. The diagnostic age differences between the younger and older siblings ranged from 5 years early to 3 years later, with the older sibling receiving a later diagnosis at an average age of 4.3 years, aligning with the previous literature [27,30,33].

Most parents found early diagnosis highly beneficial across multiple domains, including better school preparedness, earlier access to support services, increased options for clinical trial participation, more time to evaluate treatment options, absence of diagnostic odyssey, and easier access to medical assistance through state programs or Medicaid. These positive experiences align with the documented benefits of early diagnosis as evidenced in past studies and echo the sentiments expressed by families in previous research, underscoring their preference for early awareness of their children’s diagnoses as newborns [12,27,30,34]. However, some parents identified increased time to worry as a predominant harm of early diagnosis, consistent with attitudes toward NBS found within the past literature indicating higher anxiety levels among mothers of children screened for DMD [35]. Qualitative analysis revealed positive effects of early diagnosis, including parents feeling more empowered to advocate for their children’s needs, having a better understanding of available resources, and early and broader access to treatments, consistent with parental expectations in previous research [30,36,37,38]. Additionally, we observed a notable positive effect—improved health outcomes of youngest children compared to oldest, which is a significant observation. These findings emphasize the importance and efficacy of screening during the newborn period for DMD, which aligns with the criteria for including conditions in the RUSP. A few negative effects of early diagnosis, such as impacts on the parents’ ability to bond, the emotional burden associated with additional diagnosis, lost time, and the emotional and physical impacts that come with treatment side effects, also aligned with the past literature [39,40,41,42,43]. Post hoc analyses revealed significant variations in parental experiences related to early diagnosis based on key cohort differences. Parents found the domains of preparedness for expectations and having clinical trial options more beneficial when the younger child received a diagnosis by age 2 and 2–5 years earlier than the older child. Furthermore, parents found eligibility for gene therapy, ample time for evaluating treatment options, and the ability to adjust expectations more beneficial when the older and younger children were under 13 and 10 years old, respectively.

Advocates for DMD push for its inclusion on the RUSP due to the potential for more effective treatments with earlier diagnosis, benefiting males with DMD and females with dystrophinopathy [4]. The recent addition of spinal muscular atrophy (SMA) to the RUSP in 2018 has set a precedent for neuromuscular disorders to be included in NBS panels. Expert DMD care physicians [36] and affected families prefer early diagnosis for prompt access to treatments that are most effective in the newborn stage [30]. Therapies for DMD tend to be more impactful before significant muscle degeneration occurs [24,25,26]. This study highlights differences in the years of diagnosis among siblings, resulting in potential variations in the availability of key treatment options. With the FDA approval of four exon-skipping drugs between 2016 and 2021 [8], deflazacort in 2017 [6], gene therapy, Elevidys (delandistrogene moxeparvovec-rokl), the corticosteroid vamorolone in 2023 [9], and the nonsteroidal drug, Duvyzat in 2024 [10], younger children can access a broader array of treatment options earlier in their disease progression, as demonstrated in 3.0% of families, where the youngest child initiated medical therapy five years earlier.

The study findings addressed key evidence gaps in terms of benefits and harms associated with early diagnosis of a rare condition like DMD in children who undergo screening in the newborn period due to a prior diagnosis of their sibling, who are either clinically detected or screened through routine care. Traditionally, documenting these effects has been difficult due to the lack of data on individuals screened at an earlier stage compared to unscreened individuals, as DMD diagnosis typically occurs at ages 4–5. With rare conditions, the limited prevalence presents challenges in conducting large-scale studies to detect meaningful differences between screened and unscreened individuals. Our unique study design overcomes this limitation by focusing on families with multiple children diagnosed with DMD. This allows us to demonstrate earlier treatment initiation and improved health outcomes in siblings diagnosed during the newborn period compared to those diagnosed later. This novel research not only bridges the evidence gap regarding the efficacy of screening during the newborn period but also addresses the lack of sibling data. Our research holds significant potential to influence the consideration of DMD for inclusion in the RUSP NBS panel.

It is important to note that this study does not address other key considerations for DMD inclusion in the RUSP NBS panel, which include evaluating the characteristics of screening tests and the availability of confirmatory tests. We are unable to evaluate screening test characteristics because our study data includes information on children who are already clinically diagnosed with DMD. Additionally, evaluating the availability of confirmatory tests is beyond the scope of this manuscript. Furthermore, the results are subject to recall bias, as parents may not accurately remember specific details or nuances over time, potentially impacting the validity of the findings. While the study provides valuable insights into the relative importance of various aspects of early diagnosis, the potential variability in individual perceptions of experiences may add a level of uncertainty in determining the hierarchy of importance across different domains.

## 5. Conclusions

The study examined the potential benefits and harms of early diagnosis of Duchenne muscular dystrophy among children screened during the newborn period based on prior diagnosis of their siblings in rare cases of families having multiple children with DMD. Most parents reported experiencing net benefits from early diagnosis. Many reported that early diagnosis may be associated with worry, and a smaller group of parents reported mixed emotions. Earlier diagnosis helps in making major life decisions, which are significant during the critical phases of the disease and treatment journey for children with DMD and their caregivers. The insights gained from understanding families’ perspectives on their lived experiences, particularly in cases of an early diagnosis for one child due to the diagnosis of another, have the potential to reshape the landscape of newborn screening.

## Figures and Tables

**Figure 1 IJNS-10-00032-f001:**
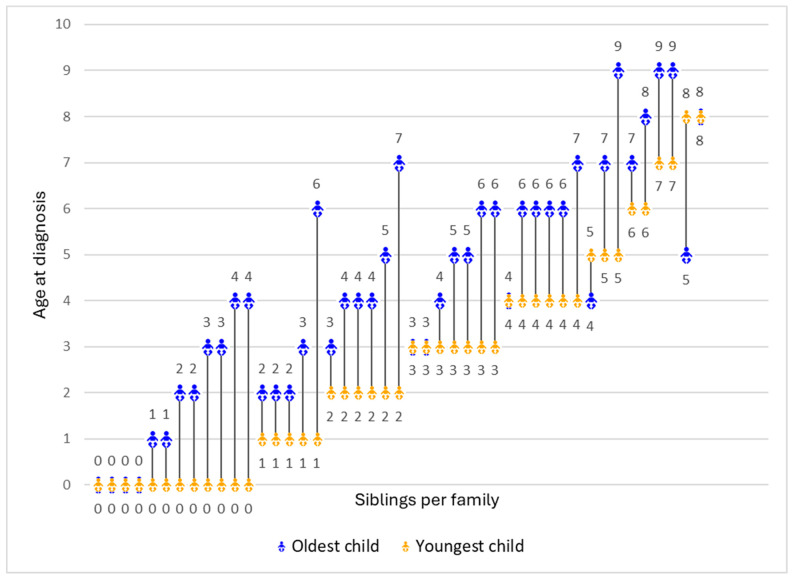
Age of diagnosis for the oldest and youngest child in each family (*N* = 45). Each line represents the difference in the ages of diagnosis between the youngest (represented in orange) and the oldest child (represented in blue) per family.

**Figure 2 IJNS-10-00032-f002:**
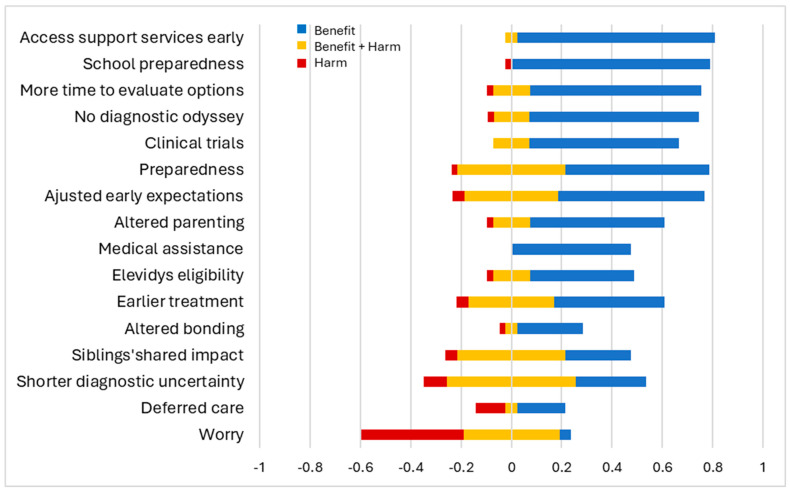
Parental view on early diagnosis. The x-axis represents the standardized scores of experience items across the bidirectional scale. The figure shows if parents perceived their lived experience as benefits (in blue), harms (in red), or both benefits and harms (in yellow).

**Figure 3 IJNS-10-00032-f003:**
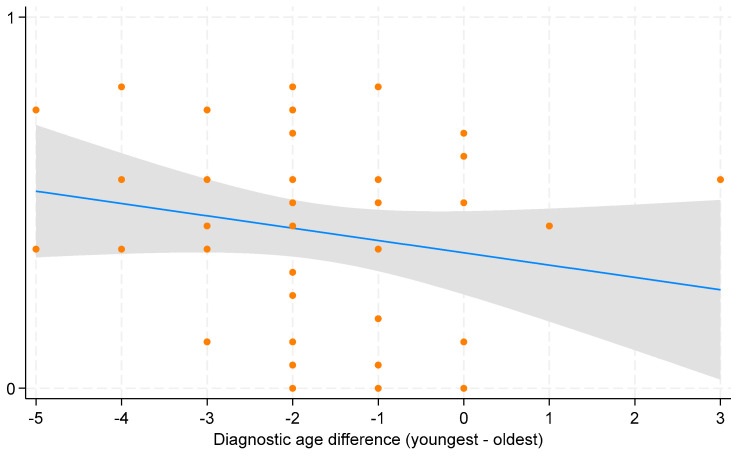
Scatterplot with fitted line for average parental experience of early diagnosis. The x-axis shows the differences in diagnostic age between the youngest and oldest child per family, with negative values indicating earlier diagnosis for the youngest child and positive values indicating later diagnosis compared to the oldest child. The scatterplot displays individual data points for each observation of average parental experience across diagnostic age difference. The blue line represents the fitted line indicating the estimated change in average parental experience for a one-unit change in diagnostic age difference. The grey bar around the blue line represents the 95% confidence interval (CI).

**Table 1 IJNS-10-00032-t001:** Descriptive statistics.

Characteristics of Parents (*N* = 45)	*n* (%)		
Age *	**43.5 (32–60)**		
White	35 (77.8)		
Black	1 (2.2)		
Hispanic/Latino	4 (8.9)		
Asian	3 (6.7)		
American Indian/Alaska Native	1 (2.2)		
**Characteristics of children (*N* = 45 each)**	**Oldest, n (%)**	**Youngest, n (%)**	***p*-value**
Age at diagnosis *	4.3 (0–9)	2.6 (0–8)	<0.001
Current age *	13.6 (2–28)	10.8 (1–26)	<0.001
Loss of ambulation ^†^	22 (62.9)	12 (38.7)	0.003
Wheelchair dependence age *	10.3 (8–17)	10.4 (8–13)	0.838
Medical therapy	41 (91.1)	38 (84.4)	0.183
Medical therapy starting age *	6.9 (1–20)	5.7 (0–19)	<0.001
Clinical trial	14 (31.1)	13 (30.2)	0.660
Clinical trial starting age *	7.6 (1–12)	6.8 (0–9)	0.396
Early Intervention Service (EIS)	23 (52.3)	25 (58.1)	<0.001
EIS starting age *	1.8 (0–3)	1.4 (0–3)	0.138
Individualized Educational Plan (IEP)	37 (82.2)	34 (79.1)	0.660
IEP starting age *	5.9 (3–10)	5.6 (3–17)	0.263

* All age variables (in years) are reported as the mean, with range in parenthesis. † The question on ambulatory status was asked only if the current ages of children with DMD were greater than 10 years.

**Table 2 IJNS-10-00032-t002:** Positive effects of early diagnosis.

Theme	Subtheme	Illustrative Quote
Response	Diagnostic considerations	“…the benefits far outweigh the drawbacks…” (Parent 34)
	Advocating needs	“Knowing my son’s diagnosis helped me to fight for her. I knew something was wrong and continued to fight even though everyone dismissed me”. (Parent 5)
	Parenting expectations	“With my younger son, I have adapted expectations based on the capabilities I learned with my older son. For example, I am quicker to help him when he is tired or on stairs”. (Parent 8)
Knowledge	System navigation	“I would always know what to advocate for since I had just gone through it with my older son. For example, requesting an aide at the right time, navigating the playground, recess supervision, and classroom modifications as they got older and needed to use their chairs in middle school/high school”. (Parent 22)
	Education resources	“We knew what to expect when dealing with a school, what resources we actually had, knowing the lines of legality, like when a school principal told my son he had to use the stairs, when it was clearly stated in his IEP and 504 that he was specifically not allowed to use the stairs”. (Parent 17)
	Disease characteristics	“We knew why and understood why our youngest wasn’t able to do age typical things”. (Parent 14)
Planning	Systematic connections	“Having everything in place for my [son’s] care. Doctor referrals ahead of time, got equipment needed when it was needed with no issues. Got their Medicaid set up as soon as we received diagnosis”. (Parent 16)
	School accommodations	“Having a team at school that not only understands their condition but fully supports them has been critical for their success”. (Parent 34)
	Parenting expectations	“… allowed us to change our expectations of the boys and be excited for the little gains in independence”. (Parent 43)
Treatment	Early treatment	“More treatments seem to be available at a younger age…” (Parent 39)
	Clinical trials	“… able to participate in clinical trials that have allowed them to extend their mobility”. (Parent 18)
	Available therapies	“Gene therapy has given my children strength and the opportunity to enjoy playing with each other and peers”. (Parent 43)
Health	Disease progression	“… my youngest has the abilities to do things that my oldest wasn’t able to from his delayed diagnosis”. (Parent 39)
	Prolonged ambulation	“The strength in his legs has improved and also his gross motor skills improved. His walking and running and climbing stairs improved quite a lot”. (Parent 38)
	Longer life	“Good heart and lungs function”. (Parent 42)

**Table 3 IJNS-10-00032-t003:** Negative effects of early diagnosis.

Theme	Subtheme	Illustrative Quote
Life impacts	Emotional burden	“Finding out about their diagnosis pushed me into a state of high anxiety and impacted my ability to be full there for them”. (Parent 43)“… it has… an emotional toll, especially seeing the side effects from the medicine”. (Parent 21)
	Lost time	“[Putting] too much emphasis on Duchenne and not allowing it to be just one part of the child’s life”. (Parent 27)“I wanted my boys to have a life to enjoy…” (Parent 16)
Treatment	Behavioral impacts	“We stopped prednisone…didn’t like the side effects, such as mood swings. [It] wasn’t worth the extra issues we would have to deal with, on top of the Duchenne”. (Parent 16)
	Side effects	“The harm would be the side effects of the medication. Medication and doctor’s appointments/tests become almost become your life”. (Parent 21)“There are pros and cons for earlier treatments…” (Parent 34)

## Data Availability

The data supporting the study findings are not publicly available due to privacy restrictions. The data are available on request from the corresponding author.

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
