# Peer review of "Assessing the Benefits and Harms Associated with Early Diagnosis from the Perspective of Parents with Multiple Children Diagnosed with Duchenne Muscular Dystrophy"

_2409-515X, 2024, doi:10.3390/ijns10020032_

Round 1

Reviewer 1 Report

Comments and Suggestions for Authors

I think this was a well-written manuscript.  It was interesting that 84.4% of younger DMD siblings initiated DMD-appropriate therapy compared to 91.1 % (p=0.183) but significantly more younger sibs with DMD started medical therapy earlier (p<0.001) from Table 1. From the same table 1 therefore, p-value quoted appears to have been p<0.001 instead of p<0.01 in the manuscript line 139.   The manuscript  may help with RUSP consideration of DMD to the NBS panel.

Reviewer 2 Report

Comments and Suggestions for Authors

This manuscript describes a study of the attitudes of parents who have two or more children with DMD with respect to the value of the earlier diagnosis of the second affected sibling compared to the first.  The research is timely, well-conducted, and effectively presented.  The issue of newborn screening for DMD is important.

My only recommendation for the authors is to more closely align the results with the 2023 report of the Advisory Committee one Heritable Disorders in Newborns and Children Nomination and Prioritization Workgroup on DMD.  (https://www.hrsa.gov/sites/default/files/hrsa/advisory-committees/heritable-disorders/meetings/dmd-nomination-summary.pdf)

This report did not recommend moving forward with a full review of DMD for several reasons, although primarily due to limitations of the screening and confirmatory tests.  An additional key issue was questions over clinical utility and the Workgroup noted the relative lack of research documenting the benefits of early detection, including the lack of sibling studies.  

My recommendation for the authors is to highlight this element of the report and the degree to which this study helps address the important issue of benefit.  The authors also might highlight the degree of difficulty in demonstrating benefit/harm between screened and unscreened children at a population level for rare conditions. These difficulties make sibling studies particularly valuable. However, the authors should also note that there are other critical issues regarding screening and confirmatory testing that remain to be solved, at least according to the Advisory Committee.  That is, the paper has a tone of advocacy for newborn screening for DMD but does not touch on the full range of considerations.
